# Decompressive Craniectomy Improves QTc Interval in Traumatic Brain Injury Patients

**DOI:** 10.3390/ijerph17228653

**Published:** 2020-11-21

**Authors:** Wojciech Dabrowski, Dorota Siwicka-Gieroba, Chiara Robba, Rafael Badenes, Katarzyna Kotfis, Todd T. Schlegel, Andrzej Jaroszynski

**Affiliations:** 1Department of Anaesthesiology and Intensive Care, Medical University of Lublin, 20-954 Lublin, Poland; dsiw@wp.pl; 2Department of Anaesthesia and Intensive Care, Policlinico San Martino, 1100 Genova, Italy; kiarobba@gmail.com; 3Department of Anaesthesiology and Intensive Care, Hospital Clìnico Universitario de Valencia, University of Valencia, 46010 Valencia, Spain; rafaelbadenes@gmail.com; 4Department Anaesthesiology, Intensive Therapy and Acute Intoxications, Pomeranian Medical University, 70-111 Szczecin, Poland; katarzyna.kotfis@pum.edu.pl; 5Department of Molecular Medicine and Surgery, Karolinska Institute, SE-171 76 Stockholm, Sweden; ttschlegel@gmail.com; 6Nicollier-Schlegel SARL, 1270 Trélex, Switzerland; 7Department of Nephrology, Collegium Medicum, Jan Kochanowski University of Kielce, 25-736 Kielce, Poland; jaroszynskiaj@interia.pl

**Keywords:** traumatic brain injury, cardiac disorders, electrocardiography, cardiac arrhythmias, the index of cardio-electrophysiological balance

## Abstract

Background: Traumatic brain injury (TBI) is commonly associated with cardiac dysfunction, which may be reflected by abnormal electrocardiograms (ECG) and/or contractility. TBI-related cardiac disorders depend on the type of cerebral injury, the region of brain damage and the severity of the intracranial hypertension. Decompressive craniectomy (DC) is commonly used to reduce intra-cranial hypertension (ICH). Although DC decreases ICH rapidly, its effect on ECG has not been systematically studied. The aim of this study was to analyze the changes in ECG in patients undergoing DC. Methods: Adult patients without previously known cardiac diseases treated for isolated TBI with DC were studied. ECG variables, such as: spatial QRS-T angle (spQRS-T), corrected QT interval (QTc), QRS and T axes (QRS_ax_ and T_ax_, respectively), STJ segment and the index of cardio-electrophysiological balance (iCEB) were analyzed before DC and at 12–24 h after DC. Changes in ECG were analyzed according to the occurrence of cardiac arrhythmias and 28-day mortality. Results: 48 patients (17 female and 31 male) aged 18–64 were studied. Intra-cranial pressure correlated with QTc before DC (*p* < 0.01, *r* = 0.49). DC reduced spQRS-T (*p* < 0.001) and QTc interval (*p* < 0.01), increased Tax (*p* < 0.01) and changed STJ in a majority of leads but did not affect QRS_ax_ and iCEB. The iCEB was relatively increased before DC in patients who eventually experienced cardiac arrhythmias after DC (*p* < 0.05). Higher post-DC iCEB was also noted in non-survivors (*p* < 0.05), although iCEB values were notably heart rate-dependent. Conclusions: ICP positively correlates with QTc interval in patients with isolated TBI, and DC for relief of ICH reduces QTc and spQRS-T. However, DC might also increase risk for life-threatening cardiac arrhythmias, especially in ICH patients with notably prolonged QTc before and increased iCEB after DC.

## 1. Introduction

Cardiac disorders are common after traumatic brain injury (TBI) and considered a consequence of so-called brain-heart crosstalk. Cardiac dysfunction occurs in nearly 50% of TBI patients and associates with poor outcome [1]. Common cardiological consequences of TBI include myocardial ischemia, diminished cardiac contractility and electrocardiographic (ECG) abnormalities including cardiac arrhythmias [2,3]. ECG abnormalities such as ST segment depression or elevation, T-wave inversion, new Q waves, prolonged PR and corrected QT (QTc) intervals occur in more than 50% of patients with severe TBI [1,2,3,4,5]. Recently, we documented a significant increase in the spatial QRS-T angle (spQRS-T) and prolongation of the QTc interval within the first 24 and 48 h after moderate or severe TBI [6]. These changes were most spectacular just prior to any subsequent foraminal herniation of the brain, which then itself significantly reduced the ECG abnormalities.

Intracranial hypertension (ICH)-related cardiac disorders were documented nearly 50 years ago [7,8]. An increase in ICP can cause changes in the T wave, PR interval and STJ segment, and prolong the QTc [5,8,9]. Prolonged QTc, longer than 470 ms, has been shown to be an independent risk factor for the development of life-threatening cardiac arrhythmias in patients with subarachnoid hemorrhage [10]. There is also strong evidence that prolonged QTc (>500 ms) as well as widened spQRS-T more than 120° for men and 90° for women are risk factors for life-threatening cardiac arrhythmias and sudden cardiac death not only in TBI patients, but also in the general population [11,12,13]. Of note, prolonged QTc interval reflects disorders in cardiac repolarization and widened spQRS-T the difference between ventricular depolarization and repolarization [11,12,13]. An imbalance of cardiac depolarization reflected by QRS duration, and cardiac repolarization (QTc interval) strongly predispose to cardiac arrhythmias. In 2013, Lu and co-workers proposed a new marker for prediction of malignant cardiac arrhythmias—the index of cardio-electrophysiological balance (iCEB) [14]. It has been shown that increased values of iCEB are associated with an increased risk of the Torsade de Pointes (TdP)-induced ventricular arrhythmia in patients after hemodialysis [15]. Increased values of iCEB have also been found in patients treated for supraventricular cardiac arrhythmia [16]. However, changes in iCEB have never been studied in TBI patients.

TBI-related cardiac disorders associated with abnormal repolarization may result from stimulation/depression of the midbrain and the dorsal pontine regions, and Dorsal Vagal Motor Nucleus (DVMN) [17,18]. An increase in ICP associated with a decrease in cerebral blood flow affects cerebral activity and disturbs cardiac autonomic activity, often leading to electrocardiographic abnormalities.

Decompressive craniectomy (DC) is commonly performed to reduce intracranial hypertension (ICH) refractory to medical treatment [19,20]. It has been documented that increased ICP intensifies risk of cardiac disorders [6,7,8,9,10]. However, the effect on cardiac function of rapid decompression with consequent reduction of ICH has not been documented.

The aim of the present study was to analyze the effect of decompressive craniectomy on electrocardiographic variables in TBI patients. We hypothesized that rapid decrease in ICP and improved cerebral blood flow following DC might correct ICH-related disorders in electrocardiographic abnormalities and cardiac function.

## 2. Methods

This prospective, observational study was conducted in accordance with the Declaration of Helsinki and applicable regulatory requirements approved by the Institutional Review Board and the Bioethics Committee of Medical University at Lublin, Poland (KE-0254/26/2019). Informed consent was obtained from patients’ legal representatives, as all enrolled patients were unconscious or/and sedated at the moment of the inclusion in the study.

### 2.1. Patient Selection

Adult patients who were treated for severe isolated TBI (classified according to the Glasgow Coma Score (GCS) and Rotterdam computed tomography score (CTS) care system [21]), and who required DC due to severe cerebral edema or intracranial hematomas, were included in the study. Exclusion criteria were pregnancy, age below 18 years, patients with pre-injury cardiac diseases or cardiac/cardiosurgical history. Moreover, patients with a pre-injury history of endocrine, metabolic, pulmonary or hepatorenal diseases, as well as drug-intoxicated patients and prior transplant recipients, were excluded.

For the entire duration of the ICU stay, relevant demographic, clinical and laboratory data along with daily assessment of fluid balance, sepsis-related organ failure assessment (SOFA) score, and advanced hemodynamic monitoring variables (obtained with EV1000) were registered in an electronic database. Data on mortality was collected on day 28.

### 2.2. Patients’ Monitoring and Management

Our monitoring and treatment techniques have been previously described [22]. Arterial blood pressures and heart rate (HR) were continuously measured. Additionally, hemodynamic variables such as cardiac output/index (CO/CI), stroke volume variation (SVV), systemic vascular resistance index (SVRI) and central venous pressure (CVP) were monitored using the EV 1000 platform. Masimo Root monitor (USA) with SEDLine was used for continuous measurement of regional cerebral oxygen saturation (SrO_2_), fronto-temporal electroencephalography, peripheral saturation (SpO_2_) with hemoglobin in level and Pleth Variability Index (PVI). Intracranial pressure management was based on the latest Brain Trauma foundation guidelines [23]. Hyperosmotic therapy to reduce cerebral edema—if necessary—included the administration of boluses of 15% mannitol. This treatment was discontinued in patients with osmolality higher than 320 mOsm/kg H_2_O. Blood potassium concentration was measured five times per day and eventually corrected using continuous infusion of potassium and magnesium mixture. Fluid administration and vasopressors (norepinephrine) were titrated to obtain SrO_2_ higher than 50% and mean arterial pressure (MAP) higher than 80 mmHg.

### 2.3. Anaesthesia for Decompressive Craniectomy

Total intravenous anaesthesia (TIVA) was used for decompressive craniectomy in all patients. General anaesthesia was induced and maintained with the continuous administration of fentanyl (Polfa, Warsaw, Poland) and propofol (Diprivan, AstraZeneca, Macclesfield, UK). Muscle relaxation was obtained with a single bolus of 0.1 mg/kg body weight vecuronium bromide (Norcuron, Organon, Oss, Holand), which was titrated to maintain an adequate level of muscle relaxation. The depth of anaesthesia was controlled with Patients Sedation Index (PSI) measured by Masimo Root. Anaesthesia was maintained throughout the procedure using a mixture of air (60%) and oxygen (40%), and all subjects were ventilated using intermittent positive pressure ventilation. The following ventilation parameters were applied: tidal volume (6–8 mL/kg body weight) plateau pressure < 30 cm H_2_0, and respiration rates according to the carbon dioxide levels. Ventilatory parameters were set to maintain values of end tidal carbon dioxide (EtCO_2_) between 35–40 mmHg, oxygen reserve index (ORI) 0.0–0.1 and pulse oximetry (SpO_2_) 97–100% saturation, respectively. The mixture of gases was monitored with a SpaceLabs monitor (SpaceLabs Healthcare, OSI systems, Snoqualmie, Washington, DC, USA). All patients received continuous infusion of balanced crystalloids to maintain MAP above 80 mmHg. Intravascular volume status and fluid responsiveness were controlled with Pleth Variability Index (PVI) using the Masimo device. In patients not responding to fluid therapy, a continuous infusion of norepinephrine (Polfa, Warsaw, Poland) was used. Tachycardia was treated with the ultra-selective beta-1-blockerlandiolol (Ranrapiq, Ammomed Vienna, Austria). After surgery completion, all patients were transferred to the ICU.

### 2.4. Electrocardiography and Vectorcardiography Measurement, and Studied Time Points

Surface12-lead resting ECGs with derived vectorcardiograms (VCGs) were recorded using a Cardiax device (IMED Co Ltd., Budapest, Hungary). The recordings at each time period were automatically averaged to a single median beat, and transformed into three orthogonal leads X, Y and Z according to the inverse Dower method [24,25]. The projections of the maximum vectors of QRS and T-waves in the frontal, transverse, and left sagittal planes and on the x, y, and z axes were then obtained. Next, the value for the spQRS-T angle, automatically calculated from the maximum spatial QRS and T vectors, as well as the values of the QTc interval, QRS_ax_, T_ax,_ STJ and T_a_, were obtained directly from the Cardiax commercial software. For detection of the end of T wave, QT interval and calculation of QTc, the Cardiax software utilizes a median beat related “global QT interval” algorithm similar to that described by Xue [26]. The index of cardiac-electrophysiological balance was calculated by dividing QT by QRS (iCEB = QT/QRS) [15]. The ECG variables were analyzed before DC and at 12–24 h after DC.

### 2.5. Statistical Analysis

The Shapiro-Wilk test was used to test the normality of the distribution of the results. Means and standard deviations (SD) were calculated, and the Student’s unpaired *t*-test was used for variables with normal distribution. Categorical variables were compared using the χ^2^ and Fisher exact test, or χ^2^ with Yates correction when applicable. For variables with non-normal distribution, the Wilcoxon signed-rank, Mann-Whitney-U, Kruskal-Wallis ANOVA and post-hoc Dunnett’s multiple comparison tests were used. Linear regression analysis was performed by using the Pearson’s test for variables with normal distribution and the Spearman’s test for variables with non-normal distribution. Multiple stepwise regression analysis was performed to estimate the potential influence of various factors on changes in spQRS-T angle, QTc, QRS_ax_, T_ax_ and STJ. The following independence parameters were entered into the model: dose of norepinephrine, plasma osmolality and ICP. A *p* < 0.05 was considered significant.

## 3. Results

Sixty-two consecutive adult patients without previous cardiac history admitted to the intensive care unit for isolated TBI between January 2019 and December 2019 were considered for inclusion in the study. Fourteen underwent emergency DC just after the admission to hospital, before the admission to ICU, and those patients were excluded due to lack of sufficient cardiac evaluation. Finally, 48 patients (17 female and 31 male) aged 18–64 were included (Figure 1). Fifteen of the included patients underwent DC within 24 h after the admission to ICU, 28 between 24 and 48 h of treatment, and five on day three after admission. Intra-cranial pressure was monitored in 27 patients. The mean value of ICP just before DC was 42.11 ± 11.1 mmHg.

Mean HR before surgery was 91.6 ± 26.6 beats/min and significantly decreased to 83.6 ± 19.5 beats/min after DC (*p* < 0.05). Intra-cranial pressure correlated with QTc calculated with Bazett formula before DC (*p* < 0.01, *r* = 0.49). However, the significant correlation was lost when QTc was calculated with either the Fridericia or Framingham formulas. After decompressive craniectomy, spQRS-T was reduced, T_ax_ increased and QTc interval was shortened (Figure 2), whereas QRS_ax_ and iCEB did not change significantly (Figure 3). Additionally, the dose of norepinephrine correlated with spQRS-T before DC (*p* < 0.01, *r* = 0.39) and with QTc after DC (*p* < 0.05, *r* = 35). Plasma osmolality inversely correlated with QRS_ax_ before and after DC (*p* < 0.05, *r* = −0.37 and *p* < 0.05, *r* = −0.31, respectively). Multivariable regression showed that iCEB significantly depended on HR and dose of norepinephrine before DC (Table 1).

Changes in the STJ segment were noted in the following leads: II, III, aVF and V_6_ (Table 2). There were significant correlations between plasma osmolality and STJ in leads II, III and aVR (*p* < 0.01, *r* = −0.4; *p* < 0.05, *r* = −0.36; *p* < 0.05, *r* = 0.3 and *p* < 0.5, *r* = −0.34, respectively).

Thirteen patients (27.1%—4 female and 9 male) died due to brain death before 28 days of treatment. Two underwent DC during the 24 h after the admission to the ICU, 6 between 24 and 48 h of treatment and five between 48 and 72 h of treatment. Independent of the correction formula, the QTc was significantly prolonged before DC in non-survivors (Figure 4, *p* < 0.05). Decompressive craniectomy did not change iCEB in patients who survived 28 days, whereas iCEB increased after surgery in non-survivors (Figure 4). Changes in STJ segment were significantly greater in non-survivors before DC in leads: I, II, III, aVR, aVF, V_1_, V_4_, V_5_ and V_6_, and after DC in leads: I, II, III, aVR, aVF, V_5_ and V_6_ (Table 3). Multivariable regression showed that iCEB significantly depended on HR and dose of norepinephrine before DC in those patients who survived 28 days, whereas it depended only on HR before DC in non-survivors (Table 4).

Atrial fibrillation was noted in 10 patients within a few days after DC and was significantly more frequent in patients who did not survive the 28-days of treatment (χ^2^ = 6.93, *p* = 0.01, χ^2^ with Yates correction = 4.98, *p* < 0.05). Independent of the correction formula, QTc was significantly higher in patients with atrial fibrillation before DC (using Bazett formula: 562 ms [488, 579]—median and quartile 1 and 3, respectively vs. 464 ms [440, 514]; *p* < 0.01), as well as after DC (550 ms [441, 581] vs. 447 ms [420, 479], *p* < 0.05). Decompressive craniectomy did not differentially affect iCEB in patients with versus without atrial fibrillation, but iCEB was significantly higher before DC in patients with episodes of atrial fibrillation observed within a few days after DC (*p* < 0.05, Figure 5). The iCEB was also higher in non-survivors after DC (Figure 4). Changes in STJ segment were higher in patients with atrial fibrillation before DC in leads II, III, aVR and aVF (*p* < 0.01, respectively).

## 4. Discussion

The present study documented the effect of DC on electrocardiographic variables. Raised ICP correlated with prolonged QTc interval and emergency DC reduced the QTc and spatial QRS-T angle with an increase in T_ax_. Additionally, iCEB was higher before DC in patients with cardiac arrhythmias noted within a few days after DC. Decompressive craniectomy affected STJ segment in leads II, III, aVR and V_6_; however, changes in STJ appeared clinically unimportant. Changes in iCEB depended on HR and dose of norepinephrine.

Cardiac disorders related to the central nervous system have been studied for several years, and many pathomechanisms have been suggested to explain such relationships [3,4,5,6,26]. Each kind of brain injury induces massive catecholamine release and disturbs cardiac autonomic balance via increasing sympathetic activity [27,28]. Stimulation of the insular and subcortical regions in the brain leads to a complex cascade connected with neuroinflammation, autonomic dysfunction and rapid catecholamine release [27,28,29]. Significantly higher blood epinephrine and norepinephrine concentrations were noted in patients with posttraumatic brain swelling with diffuse axonal injury than in those treated for severe brain injury with large epidural hematoma complicated by brain stem shift without brain contusion [28]. Blood catecholamine concentrations were almost twice as high in patients with severe TBI and ICH than in moderate TBI, and closely corresponded to the severity of TBI and ICH [30]. Another clinical study documented rapid decrease in the arterial pressure following DC, which forced an increase in the dose of administrated norepinephrine to maintain the appropriate cerebral perfusion pressure [31]. Hence, it can be suggested that rapid decompression of ICH reduces blood epinephrine and norepinephrine concentrations, and these changes affect the ECG. Clinical study has documented that circulating epinephrine had a stronger effect on the QTc interval than norepinephrine [32]. In the present study the DC reduced the spatial QRS-T angle, increased T_ax_, shortened QTc interval and affected STJ segment. Therefore, we can speculate that these changes in the ECG might result from rapid changes in circulating catecholamine levels; however, these relationships should be confirmed in the further studies.

Decompressive craniectomy is commonly performed to reduce refractory ICH and improve outcome in TBI patients [33]. A significantly better outcome after DC has been documented in patients younger than 65 years, without chronic diseases, with diameter of both pupils < 4 mm, GCS > 6 points and without tracheostomy [33]. It is undeniable that several chronic diseases are more frequent in older people, increasing the risk for poor outcome in TBI patients. In the present study all patients were below 65 years. However, many chronic diseases, particularly cardiovascular, are asymptomatic and could have had an impact on poor outcome. Importantly, the timing of the DC seems to play an important role for the final outcome, because the majority of non-survivors underwent DC after 24 h of treatment. Our findings are thus in agreement with those in a meta-analysis performed Barthelemy et al., who documented a lower mortality in patients with severe TBI who underwent early DC [34]. However, the small number of non-surviving patients in our study notably limits our analysis.

Decompressive craniectomy reduced spatial QRS-T angle and increased T_ax_. Additionally, it shortened the QTc, and this effect was more spectacular in the survivors. Increased spatial QRS-T angle and QTc interval are known strong predictors of the risk for life-threatening cardiac arrhythmias and sudden cardiac death [10,11,12,13]. Increased risk of torsade de pointes and a doubling of 30-day all-cause mortality have also been noted in patients with prolonged QTc interval > 500 ms [11,35], with QTc prolongation also commonly being associated with hypokalemia and treatment with diuretics or/and antiarrhythmic drugs. Additionally, female gender, history of smoking, cardiac arrhythmias, thyroid disturbances and hypertension also correlate with prolonged QTc [36]. In the present study we documented a strong positive correlation between ICP and QTc interval. Decompressive craniectomy, which was commonly performed for reduction of ICP, shortened QTc interval, although significantly longer QTc after DC was noted in patients who did not survive. Based on these findings, ICH is likely an additional risk factor for QTc prolongation.

Spatial QRS-T angle is an ECG variable that specifically reflects the relationship between cardiac depolarization and repolarization. Its normal value is greater in men than in women but typically does not exceed 50°. Whereas modest increases in spQRS-T angle (50–100°) are common in many diseases, increases above 100° definitively increase the risk for sudden cardiac death [12,36,37]. A widened spQRS-T angle > 100° was associated with high all-cause mortality in clinical population [36,37]. In our study, 31% of the studied patients had spQRS-T angle > 100° before DC, but only one patient, who did not survive, continued to have spQRS-T angle > 100° after DC (data not shown). Hence, ICH likely affects spQRS-T angle, although unlike with QTc, we didn’t find a statistically significant correlation between ICP and spQRS-T.

The index of cardiac-electrophysiological balance measured as QT interval divided by QRS duration has been put forward as a derivative of the cardiac wavelength λ (effective refractory period × conduction velocity) [14,15]. In healthy volunteers, mean iCEB value hovers around 4, and elevated values are observed in healthy subjects with habitual cigarette smoking [38]. Additionally, changes in iCEB are associated with risk of ventricular arrhythmias [15]. The use of some medications, such as β-blockers or β_1_- and β_2_-adrenoreceptor agonists, increases iCEB [14,15]. Importantly, anesthesia with desflurane does not affect the iCEB [39]. In the present study the mean iCEB value oscillated around 4. However significantly higher values were noted in patients with cardiac arrhythmias, which typically manifested a few days after DC. Decompressive craniectomy also resulted in a 10% increase in iCEB. Additionally, episodes of atrial fibrillation were more frequent in patients who do not survive 28 days. Experimental studies have shown that 10% increases or decreases in iCAB could be a promising marker for risk of drug-induced cardiac arrhythmias [15,40]. Based on our findings, we can speculate that cerebral decompression destabilizes iCEB, reflected by post-DC elevations in iCEB. Therefore, significant variation in iCEB might be proposed as a predictive marker for risk of cardiac arrhythmia in TBI patients undergoing DC. However, this speculation should be confirmed in future studies.

### Limitations

Despite its novel findings, the present study also had several limitations. First, we did not analyze the effect of anesthesia on ECG and derived vectorcardiography variables. It has been documented that some anesthetics prolong the QTc interval and some, such as propofol, shorten it [41]. Second, we also did not analyze the effect of different doses of propofol and other medications used during anesthesia and for treatment of TBI on iCEB. Third, it seems rather obvious, per our findings, that iCEB has some degree of heart-rate dependency, meaning that iCEB, like the QT interval itself, should optimally be corrected for underlying heart rate in future studies. Fourth, although different doses of desflurane have not affected iCEB in patients in other studies undergoing gynecologic surgery, all of our patients also received other medications, which might have affected cardiac depolarization, repolarization or both. Fifth, the small numbers of patients, who did not survive the 28-day period and who had ICP measured using intracerebral sensors, notably limited our analysis. Sixth, we also did not measure circulating blood catecholamine concentrations. We found a significant correlation between the dose of norepinephrine and QTc and spQRS-T angle, but these relationships were rather weak. Nevertheless, we could speculate that changes in QTc and spQRS-T angle might result from TBI-related elevated concentrations of circulating epinephrine and norepinephrine [31]. Finally, from a methodological standpoint, we only studied automatically-generated ECG values that were not all tediously validated via secondary manual procedures. QTc was the only exception, for which we also manually implemented the different correction formulae. We also studied only one type of spQRS-T angle that was conveniently and automatically available to us via only one type of VCG-related transform, rather than all possible types of spQRS-T angles derivable from all possible types of transforms. And the type of spQRS-T angle that was utilized for convenience is not likely the scientifically optimal type [24,42].

## 5. Conclusions

The effect of DC on ECG has not been previously systematically studied. The present study showed that DC shortened QTc interval and reduced spQRS-T angle. Higher values of iCEB were also noted in patients with cardiac arrhythmias, although the increases were heart rate-dependent. Additionally, QTc interval prolongation was statistically significantly correlated with elevations in ICP. Our observations further document cardiac disorders in TBI patients without previously known cardiac diseases, and further confirm the phenomenon of brain-heart interaction. However, larger studies are required to corroborate the effects of DC on ECG findings, cardiac function and outcomes.

## Figures and Tables

**Figure 1 ijerph-17-08653-f001:**
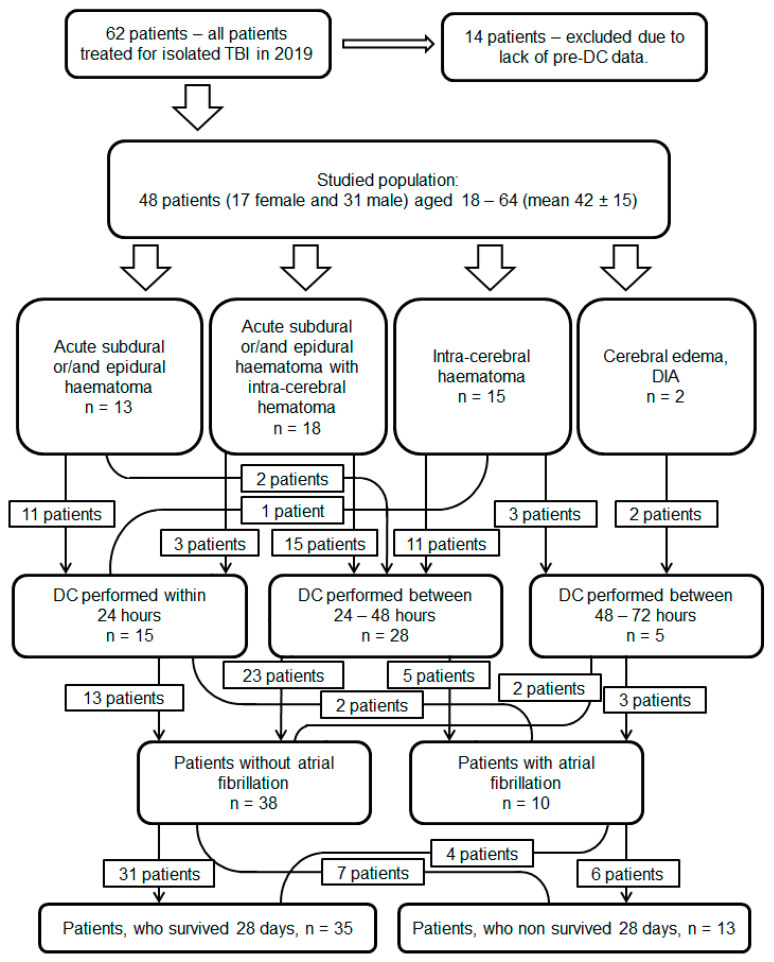
Demographic data and patient distribution.

**Figure 2 ijerph-17-08653-f002:**
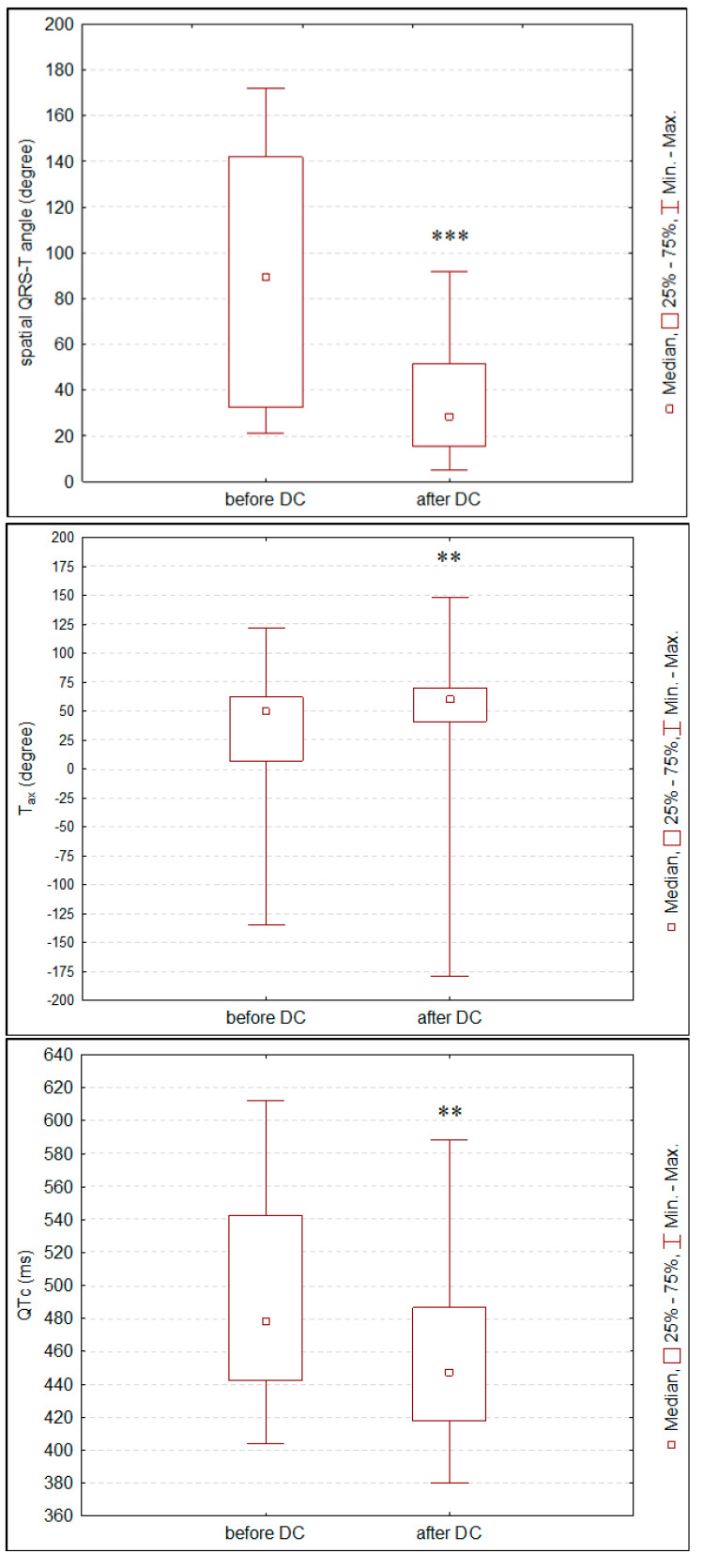
Changes in spatial QRS-T angle, T axis (T_ax_) and corrected QT (QTc) interval calculated by Bazett’s formula before and after decompressive craniectomy (DC). Significant decrease was also noted in QTc calculated with Fridericia or Framingham formulas (*p* < 0.05). ** *p* < 0.01, *** *p* < 0.001—differences between variables before and after DC (Wilcoxon test).

**Figure 3 ijerph-17-08653-f003:**
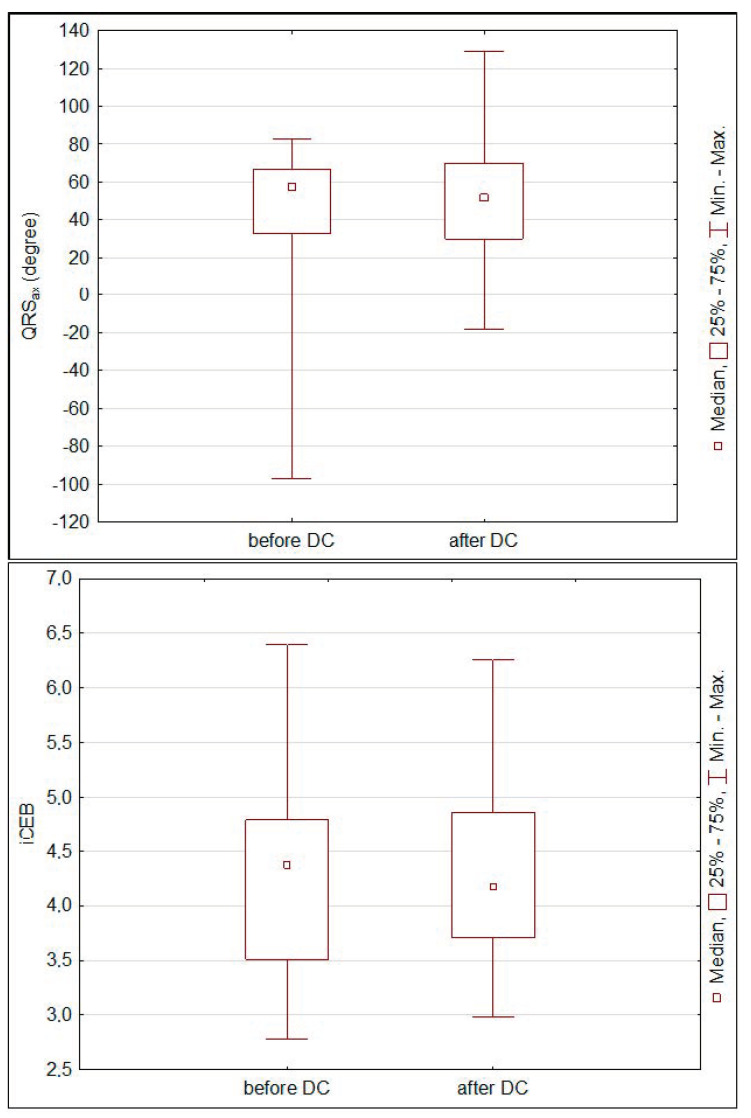
Changes in spatial QRS axis (QRS_ax_) and the index of cardio-electrophysiological balance (iCEB) before and after decompressive craniectomy (DC) (Wilcoxon test).

**Figure 4 ijerph-17-08653-f004:**
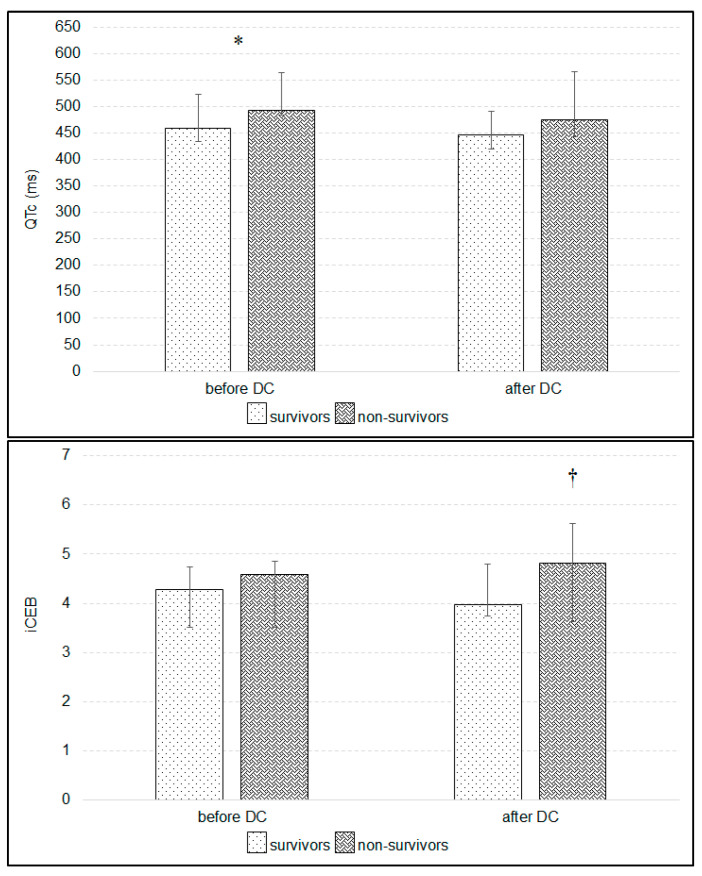
Changes in corrected QT interval (QTc) and the index of cardio-electrophysiological balance (iCEB) after compared to before decompressive craniectomy (DC) in patients who survived versus did not survive 28 days of treatment. * *p* < 0.05—differences between QTc before DC (Mann-Whitney U-test), and † *p* < 0.05—differences in iCEB after compared to before DC in patients who did not survive 28 days of treatment (Wilcoxon rank test).

**Figure 5 ijerph-17-08653-f005:**
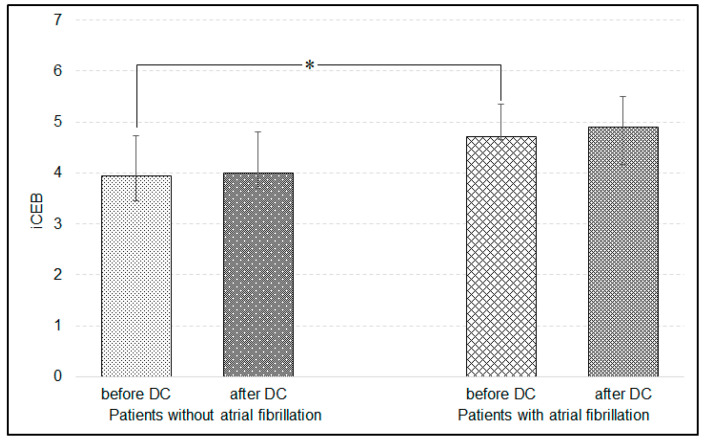
Changes in the index of cardio-electrophysiological balance (iCEB) after compared to before decompressive craniectomy (DC) in relation to episodes of atrial fibrillation observed within a few days after DC. * *p* < 0.05—difference in iCEB before DC between patients with and without atrial fibrillation (Mann-Whitney U-test).

**Table 1 ijerph-17-08653-t001:** A multiple regression analysis describing the dependence between the index of cardiac-electrophysiological balance (iCEB) and parameters selected on Pearson’s correlations in whole studied population.

*n* = 48	R = 0.61, R^2^ = 0.37 Corrected R^2^ = 0.37 F(2.45) = 13.5, *p* < 0.001					
	b *	SD of b *	b	SD of b	t	*p*
			5.99	0.366	16.343	0.0000
Heart rate	−0.633	0.123	−0.02	0.004	−5.131	0.0000
Dose of norepinephrine	0.276	0.123	0.647	0.288	2.241	0.0300

b *—standardized coefficient of multiple regression, b—non-standardized coefficient of multiple regression, SD—standard deviation.

**Table 2 ijerph-17-08653-t002:** Changes in STJ segment after compared to before decompressive craniectomy (DC).

	Before DC	After DC
I	0.01	0
	[−0.01; 0.02]	[−0.01; 0.02]
II	0	0.02 *
	[−0.03; 0.02]	[−0.01; 0.04]
III	0	0.01 **
	[−0.02; 0.01]	[−0.01; 0.03]
aVR	0	−0.01
	[−0.03; 0.02]	[−0.03; 0.01]
aVL	0	0
	[−0.01; 0.01]	[−0.01; 0.01]
aVF	0	0.01 **
	[−0.02; 0.02]	[−0.01; 0.04]
V1	0	0
	[−0.01; 0.01]	[−0.01; 0.01]
V2	0	0
	[−0.01; 0.02]	[−0.02; 0.02]
V3	0.01	0.01
	[−0.01; 0.04]	[−0.02; 0.03]
V4	−0.01	−0.01
	[−0.05; 0.03]	[−0.03; 0.03]
V5	−0.01	0
	[−0.03; 0.03]	[−0.02; 0.04]
V6	0	0.01 *
	[−0.02; 0.03]	[−0.01; 0.05]

* *p* < 0.05, ** *p* < 0.01—differences between variables before and after DC (Wilcoxon rang test).

**Table 3 ijerph-17-08653-t003:** Changes in STJ segment after compared to before decompressive craniectomy (DC) in patients who survived versus did not survive 28 days of treatment (Mann-Whitney U-test).

Lead	Time Point	Survivors	Non-Survivors	*p* Value
I	before DC	0.01	−0.01	0.005
		[0; 0.02]	[−0.01; 0]	
	after DC	0.01	−0.01	0.020
		[0; 0.02]	[−0.01; 0]	
II	before DC	0.01	−0.03	0.001
		[−0.01; 0.04]	[−0.05; 0.01]	
	after DC	0.02	−0.01	0.007
		0.01; 0.05]	[−0.04; 0.01]	
III	before DC	0	−0.01	0.016
		[−0.01; 0.02]	[−0.04; 0]	
	after DC	0.01	−0.01	0.025
		[0; 0.03]	[−0.03; 0.01]	
aVR	before DC	0	0.03	0.006
		[−0.04; 0.01]	[0; 0.05]	
	after DC	−0.02	0	0.023
		[−0.03; 0]	[−0.01; 0.02]	
aVL	before DC	0	0	0.533
		[−0.01; 0.01]	[0; 0.01]	
	after DC	0	0.01	0.339
		[−0.01; 0.01]	[0; 0.01]	
aVF	before DC	0.01	−0.02	0.004
		[−0.01; 0.03]	[−0.04; 0]	
	after DC	0.02	−0.01	0.036
		[0.01; 0.04]	[−0.03; 0.02]	
V_1_	before DC	−0.01	0.01	0.032
		[−0.02; 0.02]	[0; 0.03]	
	after DC	0	0.01	0.115
		[−0.01; 0.02]	[0; 0.03]	
V_2_	before DC	0	0	0.787
		[−0.01; 0.02]	[−0.02; 0.03]	
	after DC	0	0.01	0.444
		[−0.02; 0.02]	[−0.01; 0.02]	
V_3_	before DC	0.01	0	0.166
		[0; 0.04	[−0.03; 0.02]	
	after DC	0.01 *	0	0.390
		[−0.02; 0.03	[−0.03; 0.03]	
V_4_	before DC	0	−0.05	0.012
		[−0.03; 0.04]	−0.09; 0.02]	
	after DC	0	−0.03	0.139
		[−0.03; 0.03]	[−0.06; 0.01]	
V_5_	before DC	0.01	−0.02	0.008
		[−0.02; 0.04]	[−0.06; −0.01]	
	after DC	0.01	−0.03	0.015
		[−0.01; 0.05]	[−0.05; 0]	
V_6_	before DC	0.01	−0.01	0.008
		[−0.01; 0.04]	[−0.04; 0.01]	
	after DC	0.02	−0.02	0.003
		[0.01; 0.05]	[−0.04; 0]	

* *p* < 0.05—differences between variables before and after DC.

**Table 4 ijerph-17-08653-t004:** Multiple regression analysis showing the dependence of the index of cardiac-electrophysiological balance (iCEB) on parameters selected based on Pearson’s correlations in survivors and non-survivors.

		**R = 0.64, R^2^ = 0.41 Corrected R^2^ = 0.38 F(2.3) = 11.2 *p* < 0.001**					
		**b ***	**SD of b ***	**b**	**SD of b**	**t**	***p***
survivors				6.136	0.435	14.078	0.0000
	Heart rate	−0.666	0.142	−0.023	0.004	−4.682	0.0000
	Dose of norepinephrine	0.296	0.142	0.615	0.295	2.085	0.0451
non-survivors		**R = 0.63, R^2^ = 0.4 Corrected R^2^ = 0.28 F(2.1) = 3.4 *p* < 0.075**					
				6.152	0.785	7.836	0.0000
	Heart rate	−0.655	0.253	−0.019	0.008	−2.588	0.027
	Dose of norepinephrine	0.2323	0.2533	0.7631	0.8323	0.9168	0.3807

b *—standardized coefficient of multiple regression, b—non-standardized coefficient of multiple regression, SD—standard deviation.

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
