# Peer review of "Decompressive Craniectomy Improves QTc Interval in Traumatic Brain Injury Patients"

_ijerph, 2020, doi:10.3390/ijerph17228653_

Round 1
Reviewer 1 Report
The article describes ECG changes following decompressive craniectomy performed for the relief after traumatic brain injury.
The validity and value of the study described in the present version of the article cannot be adequately reviewed because crucial information is missing. In particular:
- What were the heart rate changes associated with decompressive craniectomy?
- How was the end of the T wave determined? Was any visual review (e.g. by a certified cardiologist) used to validate the QT interval measurements?
- The same question on the QRS complex measurement.
- The authors report the QRS-T angle calculated by the least sophisticated method. Are the same results obtained when using more robust measurements (e.g. by the XYZ areas of the QTS complex and T waves)?
- How much are the QTc results dependent on Bazett’s correction? (E.g. would the same results be obtained with Fridericia or Framingham corrections?)
- Why do the authors believe that the iCEB index is heart rate independent? (No heart rate correction of the index was used.)
Author Response
Dear Reviewers
We thank the reviewers for their very helpful reviews. Based on the reviewers' suggestions we have now revised the manuscript and marked all changes in the text in red. Responses to the reviewers' comments are below.
Sincerely,
Wojciech Dabrowski
Reviewer 1
- What were the heart rate changes associated with decompressive craniectomy? Thanks, we’ve now added the changes in HR to the Results section (lines 165-166).
- How was the end of the T wave determined? Was any visual review (e.g. by a certified cardiologist) used to validate the QT interval measurements? The same question on the QRS complex measurement. The QRS and QT intervals and end of the T wave were determined automatically by the Cardiax software. For QT interval calculation, the software uses the median of beats with tangential methods based on derivatives in a manner similar to that described by Xue JQ, Comput Cardiol 2006,33:385. We also performed ECG 3 times per time point and used the median (or when necessary, the average) result while also excluding any obviously visually inconsistent outliers from the final result, those being very few in number. But as a result of the reviewer’s helpful comment, we have now added a related new sentence to the Methods section (lines: 139 – 141) referencing Xue (new reference 26) that will hopefully add some further clarity. Moreover, we’ve also added some material to our Limitations section regarding the study’s lack of meticulous, secondary manual ECG measurements.
- The authors report the QRS-T angle calculated by the least sophisticated method. Are the same results obtained when using more robust measurements (e.g. by the XYZ areas of the QTS complex and T waves)?
Through the use of additional, secondary advanced ECG software, we are capable of measuring the spatial “mean” QRS-T angles (that use the areas as noted) as well as other types of “spatial QRS-T angles” (e.g., those derived from singular value decomposition) in addition to the spatial “peaks” QRS-T angles that Cardiax has measured for us more directly. However, both for the sake of simplicity in presentation, and because measurements of spatial “mean” QRS-T angles do not always necessarily represent more “robust” information clinically speaking (for example the peaks angle arguably has just as much if not more clinical utility than the mean angle for detecting conditions like hypertrophic cardiomyopathy and “LVH”, cf Brown and Schlegel, Journal of Electrocardiology 44 (2011) 404–409, and study #3 in this recent doctoral dissertation:
https://openarchive.ki.se/xmlui/handle/10616/47259 (click "View/Open: Thesis (8.179Mb)" to access full PDF dissertation), we decided that studying all such possible angles was not especially critical for the present effort. Moreover, the 12-lead-to-VCG transform implemented by Cardiax (the inverse Dower) is of course also often less optimal than other transforms, for example that of Kors et al. But based on the reviewer’s helpful comment, we have also now made reference to these additional potential limitations within our Limitations section.
- How much are the QTc results dependent on Bazett’s correction? (E.g. would the same results be obtained with Fridericia or Framingham corrections?). We adopted Bazett’s formula for our study both because much of the related, previous literature has used it, and (regrettably or not) because it’s likely to still be the most commonly encountered automated formula for QTc correction utilized by busy clinicians going forward into the immediate future. We do know that one disadvantage of Bazett’s method is the fact that QTc interval is overestimated at high and underestimated at low heart rates. And that formulas like Fridericia or Framingham and Hodges are better for slow heart rates, etc. However it should also be understood that only four patients in our study had heart rates below 60 /min. We have therefore made the judgment that automatically estimated, Bazett-corrected QTc intervals were adequate for this study.
Why do the authors believe that the iCEB index is heart rate independent? (No heart rate correction of the index was used.) Thanks for this comment. We ourselves do not hold to such a belief. However, with that being stated, it would also be untrue to suggest that iCEB is so heart rate dependent that it’s rendered utterly useless as a measure without a heart rate correction. (One could, for example, attempt to posit the same concept with respect to the T-wave axis or spatial QRS-T angle, those too having some degree of heart rate dependency, although rarely if ever have they been heart-rate corrected in the scientific literature). Increased values of iCEB were previously noted in patients with ventricular tachycardia (Robyns et al [17]). And we ourselves have also noted that increased iCEB present in patients who experienced (tachy)arrhythmias a few days after DC. However the absolute levels of, as well as changes in, iCEB were comparable in patients with versus without atrial fibrillation before the DC. And significantly higher values of iCEB were only noted after DC in patients who later experienced atrial fibrillation. So while we believe that iCEB does have a HR-related dependency, destabilisation of a non HR-corrected iCEB can still be a prognostic factor for tachyarrhythmias. Nonetheless, understanding the reviewer’s helpful point again, we’ve added another statement to our Limitations section about iCEB likely being heart rate-dependent to some extent, while moreover adding another sentence to Results section that iCEB was higher in non-survivors after DC

Reviewer 2 Report
Review on:
Dabrowski et al.,
Decompressive craniectomy improves QTc intervals in traumatic brain injury patients
Subject: Changes of electrocardiac abnormalities, i.e. QTc interval in ECG, due to TBI and ist changes following decompressive craniectomy (DC).
Hypothesis: Alteration of ICP leads to changes of QTc via the brain-heart axis.
Protocol: Multicenter observational study using several parameters in 48 patients without previous cardiac diseases, well described and shown in fig. 1
Results: The multinational group of authors describe a correlation between ICP changes and the interval of QTc in a protocol recording several cardiac parameters before and 24 hours after DC
Presentation of results: good understandable and easy to follow, well described.
Tables (2): important for understanding the protocol, well presented
Figures (4): well presented, to follow the text in „Results“
Discussion: purely scientific discussion, no unproven hypothesis
References: not complete, but sufficient to compare data to other studies
Relevance of the manuscript for clinical practice: high
Limitation of study: Uncontrolled protocoll. It would be worthwile to compare a group of DC- patients (this manuscript) with a group of patients with similar characteristics and same parameters who will not be craniectomized.
Writing: Multiple orthographic mistakes
Author Response
Dear Reviewers
We thank the reviewers for their very helpful reviews. Based on the reviewers' suggestions we have now revised the manuscript and marked all changes in the text in red. Responses to the reviewers' comments are below.
Sincerely,
Wojciech Dabrowski
Reviewer 2
- Subject:Changes of electrocardiac abnormalities, i.e. QTc interval in ECG, due to TBI and ist changes following decompressive craniectomy (DC).
- Hypothesis:Alteration of ICP leads to changes of QTc via the brain-heart axis.
- Protocol:Multicenter observational study using several parameters in 48 patients without previous cardiac diseases, well described and shown in fig. 1
- Results:The multinational group of authors describe a correlation between ICP changes and the interval of QTc in a protocol recording several cardiac parameters before and 24 hours after DC
- Presentation of results: good understandable and easy to follow, well described.
- Tables (2):important for understanding the protocol, well presented
- Figures (4): well presented, to follow the text in „Results“
- Discussion:purely scientific discussion, no unproven hypothesis
- References:not complete, but sufficient to compare data to other studies
- Relevance of the manuscript for clinical practice: high
- Limitation of study:Uncontrolled protocoll. It would be worthwile to compare a group of DC- patients (this manuscript) with a group of patients with similar characteristics and same parameters who will not be craniectomized.
We thank the reviewer for the helpful comments. Indeed, it would be very interesting to compare all analysed variables in patients who did versus did not undergo decompressive craniectomy. However in general such a thing is (unfortunately) not ethically possible, because decompressive craniectomy is performed only in patients with critical and life-threatening intra-cranial hypertension who require it. And failure to perform DC in such cases would usually be construed as tantamount to clinical malpractice by potentially contributing to unnecessary patient deaths.
- Writing:Multiple orthographic mistakes. The two co-authors on the manuscript who are native speakers of the English language have thoroughly reviewed the manuscript and corrected (or at least attempted to correct) all spelling or grammatical mistakes. However, if any specific mistakes remain, we would of course be happy to correct those as well, in the event that they were specifically pointed out. We thank the reviewer again for the helpful comments!

Round 2
Reviewer 1 Report
I appreciate the changes that the authors made in the manuscript but the observation of not only statistically significant but also fairly substantial (8 bpm !!) heart rate decrease leads to further problems that need to be addressed.
- While it is true that Bazett formula became entrenched in clinical practice, this fact does not offer any proof of the scientific validity of the formula. There are, indeed, scores of publications showing that the formula is inappropriate in the presence of heart rate changes. It is not true that Bazett correction leads to problems only at slow heart rate. The authors probably confuse scientific reporting with day-to-day clinical practice since it is known that Bazett correction may underestimate QTc values at slow rate which is, in clinical practice, perhaps a more important problem than the QTc overestimation at increased heart rate. In scientific reporting, it is important to document that QTc changes are real and not only a consequence of erroneous correction. I do not understand why the authors do not wish to follow the previous suggestion of showing Fridericia and Framingham corrected QTc values. Is this because the results would no longer hold?
- The so-called iCEB index is only a very simple proportion between uncorrected QT interval and the QRS duration. Since QRS duration is only minimally heart rate dependent, the heart rate dependency of the iCEB index is practically the same as that of the QT interval. Hence, reporting this index (especially in the presence of significant heart rate changes) is as inappropriate as reporting (and statistically analysing) the uncorrected QT interval. Mentioning this essential problem only the limitation section does not make it to disappear. The authors should either show that the changes in the iCEB index are independent of heart rate (e.g. using multivariable regression analysis with continuous values) or should stop using such a highly problematic measure.
- The question on the QRS-T angle was not, in principle, related to the orthogonal transformation. The question was why the authors used the calculation method that was previously shown to lead to the least stable results (e.g. Hnatkova et al, Europace, 2018; and other publications).
Author Response
Dear Reviewers
We thank the reviewers for their very helpful reviews. Based on the reviewers' suggestions we have now revised the manuscript and marked all changes in the text in red. Responses to the reviewers' comments are below.
Sincerely,
Wojciech Dabrowski
Reviewer 1
1.While it is true that Bazett formula became entrenched in clinical practice, this fact does not offer any proof of the scientific validity of the formula. There are, indeed, scores of publications showing that the formula is inappropriate in the presence of heart rate changes. It is not true that Bazett correction leads to problems only at slow heart rate. The authors probably confuse scientific reporting with day-to-day clinical practice since it is known that Bazett correction may underestimate QTc values at slow rate which is, in clinical practice, perhaps a more important problem than the QTc overestimation at increased heart rate. In scientific reporting, it is important to document that QTc changes are real and not only a consequence of erroneous correction. I do not understand why the authors do not wish to follow the previous suggestion of showing Fridericia and Framingham corrected QTc values. Is this because the results would no longer hold?
Thanks. Based on the reviewer’s comment, we have now also additionally calculated QTc values based on the Fridericia and Framingham formulas. And indeed, when using those formulas, the statistically significant correlation between ICP and QTc (when using Bazett’s) is lost. However the rest of the QTc-related results, arguably the most relevant and important ones, did not change. All of this has now been outlined with some detail in the further revised text. We thank the reviewer for this helpful comment.
2. The so-called iCEB index is only a very simple proportion between uncorrected QT interval and the QRS duration. Since QRS duration is only minimally heart rate dependent, the heart rate dependency of the iCEB index is practically the same as that of the QT interval. Hence, reporting this index (especially in the presence of significant heart rate changes) is as inappropriate as reporting (and statistically analysing) the uncorrected QT interval. Mentioning this essential problem only the limitation section does not make it to disappear. The authors should either show that the changes in the iCEB index are independent of heart rate (e.g. using multivariable regression analysis with continuous values) or should stop using such a highly problematic measure.
Thanks. We realize that as relatively new parameter within the scientific literature, that the limitation profile of the iCEB (as currently formulated) has hardly been optimally explored. However, given the pre-existing clinical literature relating to it, our desire is to just describe its changes under the conditions of our particular study while also pointing out its limitations as currently constructed, especially for the attention of those who have already published papers utilizing the parameter. And indeed, per the reviewer’s helpful comment, we’ve also further studied the issue and found significant correlations between iCEB and HR, as expected, as well as between iCEB and dose of norepinephrine before DC in the whole studied population as well as in survivors, i.e., whereas iCEB correlated only with HR in non-survivors, per our new Table 4. We have now noted all these facts with some detail in the further-revised text, while especially pointing out the heart rate dependency of iCEB (as currently constructed) in several places. We thank the reviewer for this helpful comment.
3. The question on the QRS-T angle was not, in principle, related to the orthogonal transformation. The question was why the authors used the calculation method that was previously shown to lead to the least stable results (e.g. Hnatkova et al, Europace, 2018; and other publications).
Thanks. The utilized spatial QRS-T angle calculation was one of convenience, as it is only that specific angle that (at the present time) is automatically by the Cardiax commercial software. The potential limitation of utilizing this angle from this particular method, rather than other spatial QRS-T angles derived from other methods that might produce more reliable results, has been noted in the Limitations section.
Sincerely Yours
Wojciech Dabrowski on behalf of all co-authors
